# Evaluating robustness of tabular models under meta-features based shifts

**Irina Deeva**
AI Institute
ITMO University
Saint-Petersburg, Russia
iriny.deeva@gmail.com

**Nargiza Amerkhanova**
AI Institute
ITMO University
Saint-Petersburg, Russia
471673@edu.itmo.ru

**Alena Kropacheva**
AI Institute
ITMO University
Saint-Petersburg, Russia
al.kropach@gmail.com

## Abstract

Machine learning models for tabular data often encounter distribution shifts after deployment, yet target OOD samples are frequently unavailable at evaluation time. We propose a principled protocol that leverages aggregate dataset meta-features (MFs) to construct useful proxy OOD tests from in-distribution data. Our approach has two complementary branches: (1) an MFs based splitting procedure that searches for train/test partitions which maximize differences in selected meta-features, and (2) an MFs based synthetic data generator that uses multi-objective evolutionary optimization to produce datasets whose meta-characteristics match a (possibly unavailable) target. Evaluations on real-world source/target pairs of datasets and a diverse set of learners show that MFs based splits create substantially larger distributional differences than random splits and often yield more realistic stress tests; when splits fail to predict true OOD performance, targeted synthetic generation closes the gap. Our results indicate that selected meta-features - especially mutual information, class concentration, and joint entropy - are effective signals of concept shifts and can be used to construct practical pre-deployment OOD evaluations for tabular models.

## 1 Introduction

Machine learning systems deployed in the wild routinely face data that differ from the distributions seen during training. Measuring how model performance degrades under such distribution shift - and doing so before the shifted data are observed - is a central obstacle to building reliable ML systems. In many real-world settings the true out-of-distribution (OOD) target is unavailable at evaluation time; instead, practitioners often have only partial information about the target, for example aggregate statistics or meta-characteristics (meta-features) [24] of the target dataset. This paper asks a practical question that sits at the intersection of robustness and trustworthy evaluation: *can we use meta-features (MFs) to construct test data that meaningfully predict a model's performance on an unseen OOD target?*

Given the prevalence of tabular data in many high-stakes domains, models trained on such data require systematic and rigorous evaluation procedures. OOD evaluation for tabular models exhibits distinct challenges: shifts may affect heterogeneous feature types (continuous, ordinal, categorical) and are often difficult to interpret semantically compared to visual modalities. Consequently, widely used tabular repositories and benchmarks (e.g., UCI[13], OpenML[1], TabArena[4]) typically lack mechanisms for constructing or separating datasets according to well-defined distributional shifts. Specialized resources - such as Tableshift[6], Wild-Tab[14], and TabRed[25] - partially address this gap, but remain scarce and are often limited in the range of tasks they support or in the kinds of shift they represent. These limitations motivate a principled, reproducible OOD evaluation protocol

39th Conference on Neural Information Processing Systems (NeurIPS 2025) Workshop: Reliable ML from Unreliable Data.

that enables controlled manipulation of dataset characteristics. Meta-characteristics of tabular data offer an effective solution: they are compact descriptors capturing multiple distributional axes and are sensitive to shifts affecting model behavior [15], [3]. Building on these facts, we study two complementary strategies to turn meta-features information into practical OOD evaluations. The first strategy searches for train/test splits of the available in-distribution (ID) data that maximize differences in meta-features, yielding a meta-feature split that acts as a proxy OOD test. The second strategy goes beyond re-splitting and uses an evolutionary algorithm to synthesize datasets whose meta-characteristics match those reported for a target - a path we call OOD intrinsic property characterization.

**Contributions.** (1) We formalize a principled, reproducible protocol that uses meta-characteristics to construct proxy OOD evaluations from ID data, covering both split-based and synthesis-based approaches. (2) We provide a comprehensive empirical study across datasets, models, and competing splitting methods that identifies which meta-features and combinations are most predictive of OOD model performance. (3) We show that - when splits fail - targeted synthetic data generation can close the gap between proxy and true target performance, offering a practical tool for pre-deployment evaluation when only aggregate target information is available.

The code and data can be found in the repository `https://github.com/ITMO-NSS-team/OOD_Tab_Evaluation`.

## 2 Related works

Motivated by distribution shifts that degrade performance in high-stakes tabular applications, this section reviews methods that assess robustness through standardized evaluation, split design, and synthetic stress testing. Yu et al. [27] systematize the field into three strands: (i) testing OOD performance when labeled OOD data are available, (ii) predicting OOD performance from unlabeled data, and (iii) characterizing model-intrinsic properties without access to test data. Following this classification, in this work we focus on the third setting as the most realistic. In such cases, the option exists to either rely on existing ID data or to use synthetic data to generate OOD data.

For assessing robustness under OOD, several pillars have emerged. Initially, it is possible to emphasise the notion of employing non-random splittings as a methodology for the evaluation of the quality of machine learning models in conditions of data shift. For example, researchers partition tabular datasets along time or geography (e.g. house prices by year, taxi trips by state) or along demographic attributes to create pseudo-domains. DomainBed-style protocols also use leave-one-domain-out splits (training on all but one environment) [27], [9]. However, these approaches require prior knowledge of domain divisions or shift variables, which is often unattainable in real-world scenarios. A number of works formulate the task of data splitting as a clustering task [22], [26]; however, this formulation lack interpretability regarding why performance varies. A similar idea is considered in approaches where the subpopulation with the poorest model quality is sought in the test dataset [16].

When it comes to testing models under data shift conditions using synthetic data, the concept of held-out data augmentation is worth considering [21]. This has resulted in the emergence of a new field of research focused on the synthetic transformation of test data, encompassing visual corruptions and perturbations [10], stylization [8], the addition of spurious cues [17]. In semantic segmentation, Loiseau et al. [20] show that diffusion-based generation after fine-tuning on ID data and inpainting of OOD objects yield useful test scenarios; metrics on synthetic data correlate with those on real OOD inputs, supporting the validity of such "virtual testing". Generative approaches (VAEs, GANs, diffusion models) create new data distributions or stress-test models on controlled shifts. For instance, the Bank Account Fraud (BAF) dataset was synthetically generated to introduce temporal shifts and class imbalance challenging for tabular models [11]. More recently, Puranik et al. [23] proposed TabOOD, which leverages a latent-diffusion model to generate synthetic tabular samples "at the boundaries" of the data manifold. TabOOD creates OOD-like examples (as well as minority/majority samples) to augment training, and reports large improvements in robustness under novel shifts.

It is evident from an analysis of the extant literature that the subject of evaluating the behaviour of machine learning (ML) models in the context of data shift conditions is a matter of considerable pertinence. However, it is noteworthy that there is a paucity of studies that focus specifically on the tabular domain or that seek to investigate this behaviour in an interpretable manner. The present study aims to address these two gaps.

# 3 Meta-features based shifts

## 3.1 Meta-features based splitting

Let a labeled dataset $D = \{(x_i, y_i)\}_{i=1}^{N}$ is ID dataset. Fix test size $k = \lfloor test\_size \cdot N \rfloor$. The goal is to choose a subset $T \subset \{1, ..., N\}$ with $|T| = k$ (the test indices) so that the meta-characteristics of the training set $S = \{1, ..., N\} \setminus T$ and test set $T$ differ according to user-specified criteria. Define a meta-feature extractor $M(\cdot)$ that maps a sample $A$ to a vector of meta-features values:

$$M(A) = (m_1(A), m_2(A), ..., m_p(A)). \tag{1}$$

For each meta-feature $m_j$ we define a directed distance:

$$d_j(S, T) = \begin{cases} \dfrac{m_j(S)}{m_j(T)}, & \text{if } m_j(S) \neq 0 \text{ and } m_j(T) \neq 0, \\ 0, & \text{otherwise.} \end{cases} \tag{2}$$

The selection of this ratio was driven by the objective of enhancing the manageability of the splitting process (for instance, it is desirable for the meta-characteristic to invariably exceed its value on the train in comparison to that on the test). We also define a scalar imbalance measure for a sample $A$ to obtain approximately balanced splits by class:

$$imb(A) = \frac{min_c n_c(A)}{max_c n_c(A)}, \tag{3}$$

where $n_c(A)$ is the class count of label $c$ in $A$. The imbalance objective will be then:

$$o_{imb}(S, T) = |imb(S) - imb(T) + \lambda(1 - min(imb(S), imb(T))), \tag{4}$$

with $\lambda > 0$ to penalize very imbalanced splits. We therefore obtain a multi-objective fitness vector:

$$S = \{1, \dots, N\} \setminus T, \qquad \mathbf{f}(T) = \big(d_1(S, T), d_2(S, T), \dots, d_p(S, T), o_{\text{imb}}(S, T)\big). \tag{5}$$

The search seeks Pareto-optimal $T$ that maximize the first $p$ objectives and minimize the last objective. The decision was taken to employ evolutionary algorithms as the optimisation algorithm, primarily due to the fact that not all meta-characteristics can be differentiated. The evolutionary algorithm considers the indices of the test dataset as a population individual. We employ NSGA-II for selection, which maintains a Pareto-optimal front of solutions [2]. The algorithm's overall structure corresponds to the standard steps of an evolutionary algorithm (see pseudocode 1). The mutation function replaces selected indices with unused indices from the dataset, while the crossover function exchanges unique indices between parents while preserving duplicates (see details in Appendix A, pseudocode 3). In addition to the utilisation of random generation algorithms, algorithms based on an ordered dataset were incorporated as population generation algorithms (see details in Appendix A, pseudocode 2). This modification resulted in enhanced outcomes for specific meta-characteristics. For instance, it is evident that in order to divide the sample into two parts that differ significantly in terms of their mean, it is necessary to first sort the data and then divide it. This is the reason why sorting exerts a beneficial effect on the performance of the algorithm.

## 3.2 Meta-features based generation

The second strategy involves the generation of a synthetic OOD dataset, in which the meta-features will approximate the actual values of the meta-features in the real OOD dataset. Define target meta-values $m_j^* \in \mathbb{R}^{q_j}$ (some meta-features are vectors). For a candidate synthetic dataset $S'$:

$$M(S') = (m_1(S'), m_2(S'), ..., m_p(S')), l_j(S') = ||m_j(S') - m_j^*||_2. \tag{6}$$

Multi-objective fitness (to minimize) will be then:

$$\mathbf{f}(S') = (l_1(S'), l_2(S'), ..., l_p(S')). \tag{7}$$

Evolutionary algorithms were also selected for the purpose of generating synthetic data that would minimise such fitness (Eq. 7). In the context of literature, evolutionary algorithms have not been widely employed for the generation of synthetic tabular data. However, there are notable exceptions, particularly within the domain of obtaining synthteic data with a specified level of differential privacy [19]. In this particular instance, the individual population will constitute the dataset of size $N$ itself. The general generation algorithm can be delineated as follows:

**Algorithm 1** Main: MFs based splitting

---

**Require:** Dataset $D = \{(x_i, y_i)\}_{i=1}^{N}$, test size $k$, meta-features $\mathcal{M}$, population size $P$, generations $G$

**Ensure:** Pareto front of candidate splits
 1: Population $\leftarrow \emptyset$
 2: **for** $i \leftarrow 1$ to $P$ **do**
 3:     Population$[i] \leftarrow$ CREATEINDIVIDUALBYORDERING$(D, k)$
 4:     Population$[i]$.fitness $\leftarrow$ EVALUATE(Population$[i], D, \mathcal{M})$
 5: **end for**
 6: HallOfFame $\leftarrow \emptyset$
 7: **for** $g \leftarrow 1$ to $G$ **do**
 8:     Offspring $\leftarrow$ VARAND(Population)                    ▷ apply crossover+mutation
 9:     **for all** child $\in$ Offspring **do**
10:         **if** not child.fitness.valid **then**
11:             child.fitness $\leftarrow$ EVALUATE(child, $D, \mathcal{M})$
12:         **end if**
13:     **end for**
14:     Population $\leftarrow$ SELNSGA2(Population $\cup$ Offspring, $P$) [5]
15:     HallOfFame.UPDATE(Population)
16:     record statistics (avg/min/max per objective)
17: **end for**
18: ParetoFront $\leftarrow$ extract non-dominated solutions from Population
        **return** {ParetoFront, HallOfFame, Population}

---

1. **Population generation / initialization**: sample a batch from a generative prior $G$, where $S' \leftarrow G(batch = N)$. The SOTA model for tabular data, the Forest Diffusion model [12], was selected as the generative model.

2. **Crossover**: with probability $row\_mode\_prob$ choose row-mode or column-mode; exchange $\approx 30\%$ randomly chosen rows (row-mode) or columns (column-mode) between parents (see details in Appendix A, pseudocode 7).

3. **Mutations**:
   - Add Gaussian noise to selected rows and continuous columns; scale noise per-feature (see details in Appendix A, pseudocode 4).
   - Replace selected rows by sampling continuous values from a fitted Gaussian Mixture Model and categorical by empirical category probabilities (see details in Appendix A, pseudocode 5).
   - Draw new continuous rows from multivariate normal with empirical covariance; preserve categorical by resampling (see details in Appendix A, pseudocode 6).

4. **Selection**: multi-objective selection `selNSGA3WithMemory` [5].

### 3.3 Proposed meta-features

The present study puts forward a series of five meta-features, selected via systematic sensitivity analysis to distributional shifts (see details in Appendix B), validated across synthetic scenarios (see Appendix B, Figure 3) showing consistent sensitivity to covariate and concept shifts. The Python meta-feature extractor (PyMFE) library is utilized [24], which provides standardized implementations of meta-learning features. It is important to note that ordinal characteristics (minimum, maximum) were not included in the final list of meta-characteristics, despite demonstrating a higher degree of responsiveness to shifts in the data compared to other characteristics. These characteristics were found to be too sensitive to changes in the data, which rendered optimization a challenging process.

**Information-theoretic meta-features:**

- **Mutual Information (mut_inf)** measures statistical dependence between features and the target variable. The mutual information vector is computed as

$$\mathbf{I} = [I(X_1; Y), I(X_2; Y), \ldots, I(X_d; Y)],$$

$$\text{where } I(X_i; Y) = \sum_{x_i \in X_i} \sum_{y \in Y} p(x_i, y) \log \frac{p(x_i, y)}{p(x_i)p(y)},$$

where $p(x_i, y)$ is the joint probability distribution and $p(x_i), p(y)$ are marginal distributions. For continuous variables, kernel density estimation is used to estimate probability densities.

- **Joint Entropy (joint_ent)** measures the joint entropy between features and the target variable. The joint entropy vector is computed as
$$\mathbf{H}_{joint} = [H(X_1, Y), H(X_2, Y), \dots, H(X_d, Y)],$$
$$\text{where } H(X_i, Y) = -\sum_{x_i, y} p(x_i, y) \log p(x_i, y),$$
where $p(x_i, y)$ is the joint probability distribution of feature $X_i$ and target $Y$.

- **Attribute Entropy (attr_ent)** measures the uncertainty of individual features. The entropy vector is computed as
$$\mathbf{H}_{attr} = [H(X_1), H(X_2), \dots, H(X_d)],$$
$$\text{where } H(X_i) = -\sum_{x_i} p(x_i) \log p(x_i),$$
where $p(x_i)$ is the probability of feature $X_i$ taking value $x_i$.

- **Class Concentration (class_conc)** measures the concentration coefficient for features with respect to class separation. The concentration vector is computed as
$$\mathbf{C} = [\text{conc}(X_1, Y), \text{conc}(X_2, Y), \dots, \text{conc}(X_d, Y)],$$
$$\text{where } \text{conc}(X_i, Y) = \frac{\sum_j \frac{p_{ij}^2}{p_{\cdot j}} - \sum_i p_{i\cdot}^2}{1 - \sum_i p_{i\cdot}^2},$$
where $p_{ij}$ is the joint probability of feature value $i$ and class $j$, $p_{\cdot j} = \sum_i p_{ij}$ is the marginal probability of class $j$, and $p_{i\cdot} = \sum_j p_{ij}$ is the marginal probability of feature value $i$.

**Statistical meta-features:**

- **Interquartile Range (iq_range)** measures distributional spread robustness for features. The IQR vector is computed as
$$\mathbf{IQR} = [Q_{75}^1 - Q_{25}^1, Q_{75}^2 - Q_{25}^2, \dots, Q_{75}^d - Q_{25}^d],$$
where $Q_{25}^i$ and $Q_{75}^i$ are the 25th and 75th percentiles of feature $X_i$, respectively.

Summarized meta-features are employed in this work, computed as the **means** of the individual feature vectors described above. This summarization approach reduces dimensionality while preserving the essential distributional characteristics captured by each meta-feature type.

## 4  Experimental setup

**Datasets.** We run the experiments on five source/target tabular datasets provided in our repository: `electricity`, `taxi`, `income`, `california`, and `acs_accidents`. The `taxi` and `acs_accidents` datasets originally belong to the WHYSHIFT benchmark [18], so their ID and OOD subsamples are known. The remaining three datasets were obtained from the OpenML open source repository [1], and their data shifts were modelled by dividing them according to a specific variable (for example, for `income`, this is the "gender" variable). Table 1 summarizes the key characteristics of our evaluation datasets.

**Table 1:** Dataset characteristics

| Dataset | Features | Source samples | Target samples | Classes |
|---|---|---|---|---|
| electricity | 7 | 9,987 | 10,015 | 2 |
| taxi | 8 | 10,001 | 10,001 | 2 |
| income | 13 | 20,381 | 9,783 | 2 |
| california | 8 | 10,316 | 10,320 | 2 |
| acs_accidents | 46 | 22,654 | 3,956 | 2 |

**MFs based splits (`mfs_split`).** We run multi-objective evolutionary optimization over test-index sets (test fraction 0.2) to find train/test splits maximizing meta-feature differences while maintaining class balance. Each experimental configuration repeated 5 times with different random seeds for statistical reliability. We use population size 50, 200 generations, and `selNSGA2` selection. Crossover (probability 0.7) exchanges up to 1/4 of unique indices between parents; mutation (probability 0.2) replaces indices with unused ones (Appendix A, pseudocode 3). The optimizer yields multiple solutions saved as train/test files. Models are trained on MFs based splits and evaluated on corresponding test sets using F1-score. Random splits serve as baseline.

**MFs based synthetic OOD (`mfs_synthetic`).** Given a source/target pair, we compute target meta-feature values and generate synthetic datasets matching these targets via multi-objective optimization (`selNSGA3WithMemory`). We employ population size 100, 200 generations, crossover (probability 0.5), and mutation (probability 0.16). The mutation strategy combines distribution-based sampling, Gaussian noise addition, and covariance-based generation. Experiments are repeated 5 times to account for stochastic variation. Synthetic datasets test models trained on source data, enabling OOD evaluation without target access.

**Models and evaluation.** We validate the protocol across standard learners (Logistic Regression, XGBoost) and robustness-oriented architectures (IRM, AdversarialDRO) with fixed hyperparameters and unified training/evaluation, without tuning. Logistic Regression: `max_iter=1000`, `class_weight='balanced'`, `n_jobs=-1`. XGBoost: `n_estimators=100`, `eval_metric='logloss'`, `random_state=42`. IRM trains a two-layer MLP (`hidden_size=256`, `dropout=0.1`) for 1000 iterations with invariance penalty $\lambda = 1.0$, 100 annealing iterations, Adam ($lr = 10^{-3}$), and adaptive batch size $\min(128, dataset\_size)$. AdversarialDRO: MLP trained for 10 epochs, `hidden=32`, `batch_size=64`, Adam ($lr = 0.01$), adversarial updates over label groups with $\eta\_\pi = 0.1$, $r = 0.1$, `clip_max=2.0`, $\varepsilon = 0.001$, $\beta = 0.999$ (`dropout=0.0`, `weight_decay=0.0`). Models are trained on MFs based splits or source data and evaluated on MF-split tests, real targets, or synthetic OOD datasets. Primary metric: F1-score; we report mean and variability across runs, using random splits as baseline.

## 5 Experimental results

### 5.1 Effectiveness of MFs based splits

Our primary research question asks whether meta-feature optimization can create more meaningful OOD tests than random splits. Table 2 provides direct evidence that MF_split consistently produces stronger distributional differences than random_split across most datasets and meta-feature dimensions. The bold entries in Table 2 highlight the most diverse meta-feature ratios from the neutral 1.0. By constructing a compressed representation of the data using PCA, a clear comparison was made of how the train and test sets were distributed for different splitting (figure 1). The findings unequivocally demonstrate that, in contrast to `mfs_split` splitting approach, the data exhibits minimal variation when subjected to random split.

**Table 2:** Ratios of meta-features between train and test

| Dataset | Split type | Meta-feature | | | | |
| --- | --- | --- | --- | --- | --- | --- |
| | | attr_ent | class_conc | mut_inf | iq_range | joint_ent |
| electricity | MF_split | **1.600**±0.00 | **1.909**±0.53 | **2.743**±0.74 | **1.754**±0.09 | **1.400**±0.00 |
| | random_split | 1.090±0.03 | 0.730±0.01 | 0.990±0.00 | 1.000±0.01 | 1.070±0.02 |
| income | MF_split | **1.367**±0.04 | **2.134**±0.36 | **2.147**±0.39 | **1.872**±0.10 | **1.233**±0.02 |
| | random_split | 1.050±0.00 | 0.940±0.00 | 1.010±0.00 | 1.000±0.01 | 1.030±0.01 |
| taxi | MF_split | **1.291**±0.01 | **1.300**±0.53 | **1.160**±0.08 | **1.612**±0.04 | **1.213**±0.01 |
| | random_split | 1.090±0.00 | 0.710±0.00 | 0.980±0.00 | 1.000±0.01 | 1.070±0.00 |
| california | MF_split | **1.256**±0.02 | 1.264±0.21 | **1.583**±0.55 | **1.803**±0.00 | **1.200**±0.00 |
| | random_split | 0.900±0.00 | **0.730**±0.00 | 1.010±0.00 | 0.990±0.00 | 1.100±0.00 |
| acs_accidents | MF_split | **1.369**±0.12 | **1.359**±0.29 | **1.004**±0.03 | **1.923**±0.04 | **1.291**±0.06 |
| | random_split | 1.080±0.01 | 0.980±0.00 | 1.000±0.00 | 0.980±0.02 | 1.060±0.01 |

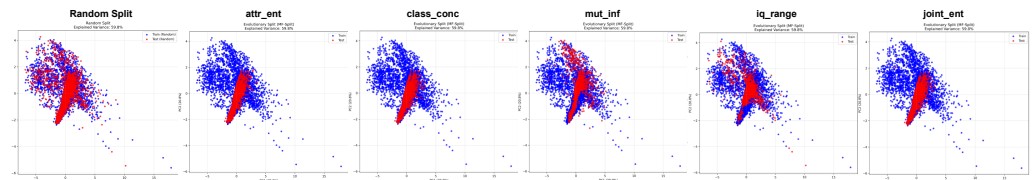

**Figure 1:** Comparison of training and test sets for the `electricity` dataset under random split and MF-split across different meta-features. The blue dots are the train, the red dots are the test.

Table 3 presents comprehensive F1-scores across all datasets, models, and meta-feature splitting criteria. For in-distribution (ID) evaluation, both training and test subsets are sampled from the source distribution using various splitting strategies. For out-of-distribution (OOD) evaluation, models are trained on the source distribution and tested on the target distribution. The colored percentages in parentheses indicate the performance difference between each model's result and the corresponding target value (on real OOD). Green values denote cases where the model performance exceeded the target (positive difference), while red values indicate performance below the target (negative difference). Bold black values show worst degradations per model-dataset pair; colored bold values indicate closest matches to target performance. The results demonstrate systematic patterns of performance degradation under MFs based splits compared to random baselines, validating our hypothesis that evolutionary optimization of meta-feature differences creates more challenging and realistic OOD test conditions.

**Table 3:** F1-scores by models across datasets and split criteria

| Metric | Dataset | LR | XGB | IRM | DRO |
|---|---|---|---|---|---|
| Random Split (ID) | electricity | $0.798 \pm 0.00$ (20%) | $0.832 \pm 0.00$ (20%) | $0.813 \pm 0.01$ (16%) | $0.814 \pm 0.02$ (17%) |
| | taxi | $0.752 \pm 0.01$ (6%) | $0.778 \pm 0.01$ (12%) | $0.790 \pm 0.02$ (15%) | $0.712 \pm 0.02$ (6%) |
| | income | $0.678 \pm 0.00$ (38%) | $0.716 \pm 0.01$ (23%) | $0.618 \pm 0.02$ (38%) | $0.514 \pm 0.04$ (10%) |
| | california | $0.823 \pm 0.01$ (8%) | $0.869 \pm 0.01$ (13%) | **0.693** $\pm 0.08$ (10%) | $0.821 \pm 0.01$ (12%) |
| | acs_accidents | $0.719 \pm 0.00$ (16%) | $0.863 \pm 0.00$ (13%) | $0.867 \pm 0.01$ (45%) | $0.702 \pm 0.07$ (22%) |
| Attr_ent (ID) | electricity | $0.813 \pm 0.00$ (22%) | $0.830 \pm 0.00$ (20%) | $0.824 \pm 0.01$ (17%) | $0.811 \pm 0.00$ (16%) |
| | taxi | $0.746 \pm 0.00$ (6%) | $0.763 \pm 0.00$ (11%) | $0.894 \pm 0.01$ (26%) | $0.724 \pm 0.01$ (7%) |
| | income | $0.628 \pm 0.00$ (33%) | $0.612 \pm 0.02$ (12%) | $0.588 \pm 0.06$ (35%) | **0.381** $\pm 0.04$ (4%) |
| | california | $0.834 \pm 0.02$ (9%) | $0.880 \pm 0.01$ (14%) | $0.953 \pm 0.01$ (16%) | $0.857 \pm 0.03$ (15%) |
| | acs_accidents | $0.542 \pm 0.10$ (2%) | $0.749 \pm 0.05$ (2%) | $0.805 \pm 0.12$ (39%) | $0.680 \pm 0.01$ (20%) |
| Joint_ent (ID) | electricity | $0.828 \pm 0.01$ (23%) | $0.841 \pm 0.00$ (21%) | $0.901 \pm 0.01$ (25%) | $0.832 \pm 0.00$ (19%) |
| | taxi | $0.751 \pm 0.01$ (6%) | $0.772 \pm 0.01$ (12%) | $0.851 \pm 0.04$ (21%) | $0.729 \pm 0.01$ (8%) |
| | income | $0.626 \pm 0.01$ (33%) | **0.605** $\pm 0.03$ **(12%)** | **0.535** $\pm 0.08$ **(30%)** | $0.392 \pm 0.01$ (3%) |
| | california | $0.877 \pm 0.01$ (13%) | $0.891 \pm 0.01$ (15%) | $0.973 \pm 0.01$ (18%) | $0.879 \pm 0.01$ (17%) |
| | acs_accidents | **0.461** $\pm 0.05$ (10%) | **0.725** $\pm 0.03$ (0%) | **0.716** $\pm 0.07$ (30%) | **0.630** $\pm 0.05$ **(15%)** |
| Mut_inf (ID) | electricity | **0.735** $\pm 0.02$ **(14%)** | **0.749** $\pm 0.01$ **(12%)** | **0.795** $\pm 0.02$ **(14%)** | **0.766** $\pm 0.01$ **(12%)** |
| | taxi | $0.723 \pm 0.01$ **(4%)** | $0.754 \pm 0.01$ (10%) | $0.899 \pm 0.01$ (26%) | $0.696 \pm 0.01$ **(4%)** |
| | income | **0.600** $\pm 0.02$ **(30%)** | $0.617 \pm 0.02$ (13%) | $0.687 \pm 0.03$ (45%) | $0.405 \pm 0.02$ (1%) |
| | california | $0.789 \pm 0.02$ (5%) | $0.866 \pm 0.04$ (13%) | $0.837 \pm 0.05$ **(4%)** | **0.786** $\pm 0.02$ **(8%)** |
| | acs_accidents | $0.718 \pm 0.01$ (16%) | $0.863 \pm 0.00$ (13%) | $0.893 \pm 0.01$ (48%) | $0.714 \pm 0.01$ (23%) |
| Class_conc (ID) | electricity | $0.736 \pm 0.01$ (14%) | $0.772 \pm 0.01$ (14%) | $0.842 \pm 0.03$ (19%) | $0.783 \pm 0.01$ (14%) |
| | taxi | **0.526** $\pm 0.10$ (16%) | **0.592** $\pm 0.07$ **(6%)** | **0.773** $\pm 0.10$ **(14%)** | **0.505** $\pm 0.10$ (15%) |
| | income | $0.622 \pm 0.01$ (32%) | $0.638 \pm 0.00$ (15%) | $0.670 \pm 0.04$ (43%) | $0.427 \pm 0.01$ **(1%)** |
| | california | **0.776** $\pm 0.01$ **(3%)** | **0.831** $\pm 0.01$ **(9%)** | $0.927 \pm 0.03$ (13%) | $0.815 \pm 0.01$ (11%) |
| | acs_accidents | $0.565 \pm 0.06$ **(0%)** | $0.785 \pm 0.03$ (6%) | $0.787 \pm 0.06$ (37%) | $0.719 \pm 0.04$ (24%) |
| IQ_range (ID) | electricity | $0.783 \pm 0.01$ (19%) | $0.806 \pm 0.01$ (17%) | $0.861 \pm 0.02$ (21%) | $0.807 \pm 0.01$ (16%) |
| | taxi | $0.741 \pm 0.01$ (5%) | $0.771 \pm 0.00$ (12%) | $0.948 \pm 0.10$ (31%) | $0.721 \pm 0.01$ (7%) |
| | income | $0.653 \pm 0.00$ (36%) | $0.685 \pm 0.02$ (20%) | $0.698 \pm 0.02$ (46%) | $0.474 \pm 0.02$ (6%) |
| | california | $0.853 \pm 0.00$ (11%) | $0.875 \pm 0.01$ (14%) | $0.876 \pm 0.01$ (8%) | $0.871 \pm 0.01$ (17%) |
| | acs_accidents | $0.671 \pm 0.00$ (11%) | $0.844 \pm 0.00$ (11%) | $0.872 \pm 0.00$ (46%) | $0.766 \pm 0.03$ (29%) |
| Target (real OOD) | electricity | 0.596 | 0.633 | 0.655 | 0.646 |
| | taxi | 0.687 | 0.655 | 0.637 | 0.654 |
| | income | 0.2974 | 0.488 | 0.240 | 0.418 |
| | california | 0.742 | 0.738 | 0.795 | 0.705 |
| | acs_accidents | 0.563 | 0.730 | 0.413 | 0.479 |

The initial observation is that the splitting of data according to characteristics such as *mut_info*, *class_conc*, and *joint_ent* frequently results in alterations in the quality of machine learning models.

This phenomenon can be attributed to the fact that these meta-features are indicative of concept shift, which is a more prevalent cause of alterations in the quality of machine learning models than covariance shift. That is why splitting by *iq_range* and *attr_ent* does not significantly change the quality of machine learning models. It is evident that all machine learning models are susceptible to the effects of splitting, even those that are considered robust. This assertion is corroborated by the conclusions drawn by the authors [7], which posit that robust models exhibit a comparable degree of quality to ERM models in the context of substantial shifts in data. It is noteworthy that although `mfs_split` generates complex splitting for machine learning models and typically compromises their quality, under certain circumstances it can be employed to enhance the quality of machine learning models in comparison to `random_split`. This phenomenon occurs when splitting by *attr_ent* meta-feature. This discrepancy may be attributed to the observation that the training sample encompasses a greater variety of observation features, characterised by higher entropy, while the test sample exhibits a higher degree of homogeneity, that is, simplicity. However, it is challenging to ascertain this with certainty, as the entropy of each predictor in the dataset differs between the train and test sets (due to the aggregation of the meta-feature vector we can't see it, see the section 3.3 for further details). This deficiency can be identified as a method's inherent shortcoming.

To contextualize the effectiveness of our MFs based split, Table 4 compares it against the MMD based clustering method proposed by Napoli and White [22]. For the Best MF row, we report the worst-case performance across all meta-features from Table 3 (indicated by bold values in that table). The percentages in parentheses show the closest gap to real OOD performance achieved by any meta-feature for each model-dataset combination. For the MMD row, we run the algorithm 5 times with different random seeds, each producing an 80%-20% train-test split. We report mean ± std across runs, with target gaps in parentheses.

**Table 4:** Performance comparison: MFs based vs. MMD based split

| Split Method | Dataset | LR | XGB | IRM | DRO |
|---|---|---|---|---|---|
| Best MF | electricity | $0.735 \pm 0.02$ **(14%)** | $0.749 \pm 0.01$ **(12%)** | $0.795 \pm 0.02$ **(14%)** | $0.766 \pm 0.01$ **(12%)** |
| | taxi | **0.526** $\pm 0.10$ (4%) | $0.592 \pm 0.07$ **(6%)** | $0.773 \pm 0.10$ (14%) | $0.505 \pm 0.10$ **(4%)** |
| | income | **0.600** $\pm 0.02$ **(30%)** | **0.605** $\pm 0.03$ **(12%)** | **0.535** $\pm 0.08$ **(30%)** | **0.381** $\pm 0.04$ **(1%)** |
| | california | **0.776** $\pm 0.01$ **(3%)** | $0.831 \pm 0.01$ (9%) | **0.837** $\pm 0.05$ **(4%)** | **0.786** $\pm 0.02$ **(8%)** |
| | acs_accidents | $0.461 \pm 0.05$ **(0%)** | $0.725 \pm 0.03$ **(0%)** | $0.716 \pm 0.07$ (30%) | $0.630 \pm 0.05$ (15%) |
| MMD | electricity | **0.419** $\pm 0.00$ (18%) | **0.335** $\pm 0.02$ (30%) | **0.435** $\pm 0.06$ (22%) | **0.355** $\pm 0.07$ (29%) |
| | taxi | $0.703 \pm 0.00$ **(2%)** | **0.564** $\pm 0.02$ (9%) | **0.690** $\pm 0.01$ **(5%)** | **0.448** $\pm 0.09$ (21%) |
| | income | $0.629 \pm 0.00$ (33%) | $0.622 \pm 0.01$ (13%) | $0.644 \pm 0.02$ (40%) | $0.650 \pm 0.01$ (23%) |
| | california | $0.828 \pm 0.00$ (9%) | **0.829** $\pm 0.02$ **(9%)** | $0.890 \pm 0.02$ (10%) | $0.855 \pm 0.01$ (15%) |
| | acs_accidents | **0.459** $\pm 0.00$ (10%) | **0.673** $\pm 0.01$ (6%) | **0.583** $\pm 0.11$ **(17%)** | **0.468** $\pm 0.12$ **(1%)** |

This evaluation demonstrates dataset-specific behavior for both methods. For electricity, MFs based split achieves gaps of 12-14% from target values, while MMD based split shows deviations of 18-30%. On taxi, MMD based split achieves tighter approximations (2-5% gaps) for most models. Both methods show substantial deviations on income (30-40%), indicating severe real-world shift. A key distinction is interpretability: MFs based split provides explicit control over distributional characteristics, while MMD uses an aggregate kernel-based distance.

### 5.2 Synthetic OOD generation effectiveness

In light of the findings from the preceding experiment, it is possible to determine certain datasets for which `mfs_split` has a significant impact on the quality of machine learning models, yet does not result in a substantial advancement in the evaluation of the actual target. The present experiment was established with the specific objective of generating synthetic OOD data for such datasets. The table 3 presents the evaluation results on synthetic OOD data for two such datasets (more detailed information on the results of synthetic OOD data generation is provided in the Appendix C). It is evident that these evaluations demonstrate a strong correlation with the actual quality of the OOD dataset. In this experiment, the number of meta-features utilised for generation was restricted, as an excessive number of meta-features resulted in suboptimal convergence. This example demonstrates the applicability of synthetic data as OOD test data. However, it may be beneficial to explore more advanced generative models for future data generation.

**Table 5:** F1-scores for models trained on original data and tested on synthetic data

| Dataset | Meta-features | LR | XGB | DRO | IRM |
|---|---|---|---|---|---|
| electricity | mut_inf, class_conc, iq_range | $0.613 \pm 0.08$ | $0.641 \pm 0.09$ | $0.587 \pm 0.08$ | $0.613 \pm 0.08$ |
| | mut_inf, class_conc | $0.611 \pm 0.01$ | $0.625 \pm 0.01$ | $0.589 \pm 0.01$ | $0.632 \pm 0.02$ |
| california | mut_inf, class_conc, iq_range | $0.636 \pm 0.05$ | $0.692 \pm 0.02$ | $0.661 \pm 0.02$ | $0.561 \pm 0.11$ |
| | mut_inf, class_conc | $0.679 \pm 0.07$ | $0.713 \pm 0.03$ | $0.628 \pm 0.10$ | $0.682 \pm 0.05$ |

## 6 Conclusion and Discussion

We introduced a practical protocol that turns dataset-level meta-features into actionable proxy OOD evaluations for tabular data, via two complementary strategies: evolutionary meta-features based splitting and meta-features based synthetic data generation. Across five real-world datasets and a set of standard and robustness-oriented learners, MFs based splits produce consistently larger distributional differences than random splits and often generate more challenging - and more realistic-stress tests for model performance. When splitting alone does not align with the true target degradation, targeted synthetic generation can close the gap, demonstrating that meta-features are useful signals for constructing pre-deployment OOD tests. Our empirical analysis identifies a small set of meta-features (notably mutual information, class concentration and joint entropy) that are particularly indicative of concept shifts and predictive of model degradation; other meta-features (e.g., simple attribute entropy or IQR) are less consistently informative for the kinds of concept shifts studied. We also highlight several limitations. First, optimization and generation are sensitive to the number and choice of meta-features: using too many objectives degrades convergence, and aggregating meta-feature vectors can hide per-predictor effects. Second, our synthetic generation relies on a set of relatively simple mutation/crossover operators and a specific generative prior; more powerful generative models (e.g., diffusion-based or conditional tabular generators) may improve fidelity. Looking forward, promising directions include: (i) integrating stronger conditional generative models to improve synthetic OOD fidelity; (ii) devising automated meta-feature selection strategies that balance informativeness and optimization tractability; (iii) extending the protocol to richer shift taxonomies (e.g., subtle covariate shifts, label noise, or compound shifts) and multi-class problems; and (iv) exploring light-weight approximations of the evolutionary search to reduce computational cost for practitioners.

## 7 Acknowledgments

This work supported by the Ministry of Economic Development of the Russian Federation (IGK 000000C313925P4C0002), agreement No139-15-2025-010.

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

# A Implementation details of the proposed algorithms

## A.1 MFs based splitting

The following pseudocodes provide a comprehensive overview of the algorithmic components employed for the purpose of data segmentation based on meta-characteristics.

---

**Algorithm 2** Create Individual By Ordering

---

**Require:** Dataset $D$ (feature matrix $X$), test size $k$
**Ensure:** Individual: list of $k$ distinct indices (test set)
1: $n \leftarrow$ number of samples in $X$
2: choose random column index $j \sim \text{Uniform}\{1, \ldots, d\}$
3: $S_{\text{sorted}} \leftarrow \text{argsort}(X[:, j])$          ▷ indices sorted by feature $j$
4: choose pattern $\in \{\text{first\_k}, \text{last\_k}, \text{contiguous}, \text{every\_other}, \text{random}\}$
5: **if** pattern = first_k **then**
6:      test_indices $\leftarrow S_{\text{sorted}}[1..k]$
7: **else if** pattern = last_k **then**
8:      test_indices $\leftarrow S_{\text{sorted}}[n - k + 1..n]$
9: **else if** pattern = contiguous **then**
10:      start $\leftarrow$ random integer in $[1, n - k + 1]$
11:      test_indices $\leftarrow S_{\text{sorted}}[\text{start}..\text{start} + k - 1]$
12: **else if** pattern = every_other **then**
13:      choose start $\in \{1, 2\}$, candidate $\leftarrow S_{\text{sorted}}[\text{start} :: 2]$
14:      test_indices $\leftarrow$ first $k$ of candidate (fallback to random if too short)
15: **else**
16:      test_indices $\leftarrow$ uniform random sample of $k$ distinct indices from $\{1, \ldots, n\}$
17: **end if**
18: **return** Individual(test_indices)

---

---

**Algorithm 3** Crossover and Mutation Operators (variation)

---

1: **function** CROSSOVER(parent$_1$, parent$_2$)
2:      $A \leftarrow \text{set}(\text{parent}_1)$; $B \leftarrow \text{set}(\text{parent}_2)$
3:      only_1 $\leftarrow A \setminus B$; only_2 $\leftarrow B \setminus A$
4:      **if** only_1 $\neq \emptyset$ and only_2 $\neq \emptyset$ **then**
5:          $m \leftarrow \min(|\text{only\_1}|, |\text{only\_2}|, \lfloor|parent|/4\rfloor)$
6:          $r \leftarrow$ uniform integer in $[1, m]$
7:          pick $r$ indices from only_1 and $r$ from only_2
8:          swap selected indices between parents preserving position uniqueness
9:      **end if**
10:      **return** child1, child2
11: **end function**
12: **function** MUTATE(individual, indpb)
13:      available $\leftarrow \{1, \ldots, N\} \setminus \text{set}(\text{individual})$
14:      **if** available $= \emptyset$ or rand() $>$ indpb **then**
15:          **return** individual
16:      **end if**
17:      $n_{\text{mut}} \leftarrow$ random integer in $[1, \min(3, |\text{individual}|)]$
18:      select $n_{\text{mut}}$ random positions in individual
19:      select $n_{\text{mut}}$ indices from available
20:      **for** each selected position $p$ and new index $i$ **do**
21:          individual$[p] \leftarrow i$
22:      **end for**
23:      **return** individual
24: **end function**

---

## A.2 MFs based generation

The following pseudocodes detail the mutation and crossover operators for generating synthetic tabular data with specified meta-characteristics. The implementation of these operators has the following characteristics:

- All mutation functions operate on a row-wise basis, considering each data point (row) independently;
- The functions handle both continuous and categorical features appropriately;
- Continuous features are modified using Gaussian noise or distribution sampling (4, 5);
- Categorical features are modified by sampling from probability distributions (6);
- The crossover function can operate on either rows or columns with configurable probabilities (7);
- All functions ensure data validity by handling NaN values and clipping categorical values to valid ranges.

---

**Algorithm 4** Mutate Noise
___

1: **function** MUTATE_NOISE(individual, mutation_prob, noise_scale, categorical_idx, continuous_idx, cat_probs, n_features)
2:     individual_data ← reshape individual to 2D array with $n\_features$ columns
3:     mutated ← copy of individual_data
4:     **for** each row $i$ in individual_data **do**
5:         **if** random value < mutation_prob **then**
6:             **if** continuous_idx exists **then**
7:                 noise_scale_adjusted ← noise_scale × |mutated[$i$][continuous_idx]|
8:                 noise ← sample from $\mathcal{N}(0, \text{noise\_scale\_adjusted})$
9:                 mutated[$i$][continuous_idx] ← mutated[$i$][continuous_idx] + noise
10:             **end if**
11:             **if** categorical_idx and cat_probs exist **then**
12:                 **for** each categorical index $j$ **do**
13:                     **if** random value < mutation_prob **then**
14:                         mutated[$i$, cat_idx] ← sample from categorical distribution with probabilities cat_probs[$j$]
15:                     **end if**
16:                 **end for**
17:             **end if**
18:         **end if**
19:     **end for**
20:     mutated ← replace NaN values with 0
21:     round and clip categorical values to valid range
22:     **return** flatten mutated array to 1D list
23: **end function**

---

**Algorithm 5** Mutate Distribution

---

1: **function** MUTATE_DIST(individual, mutation_prob, gmm, categorical_idx, continuous_idx, cat_probs, n_features)
2:     individual_data ← reshape individual to 2D array with $n\_features$ columns
3:     mutated ← copy of individual_data
4:     **for** each row $i$ in individual_data **do**
5:         **if** random value < mutation_prob **then**
6:             new_categorical ← empty list
7:             new_continuous ← empty list
8:             **if** categorical_idx and cat_probs exist **then**
9:                 **for** each probability distribution $p$ in cat_probs **do**
10:                    append sample from categorical distribution with probabilities $p$ to new_categorical
11:                **end for**
12:             **end if**
13:             **if** continuous_idx and gmm exists **then**
14:                 new_continuous ← sample from GMM
15:             **end if**
16:             new_row ← create new row with appropriate values at categorical and continuous indices
17:             mutated[$i$] ← new_row
18:         **end if**
19:     **end for**
20:     mutated ← replace NaN values with 0
21:     round and clip categorical values to valid range
22:     **return** flatten mutated array to 1D list
23: **end function**

---

---

**Algorithm 6** Mutate Covariance

---

1: **function** MUTATE_COV(individual, mutation_prob, categorical_idx, continuous_idx, cat_probs, n_features)
2:     individual_data ← reshape individual to 2D array with $n\_features$ columns
3:     mutated ← copy of individual_data
4:     **if** continuous_idx exists **then**
5:         continuous_data ← individual_data[:, continuous_idx]
6:         current_cov ← covariance matrix of continuous_data
7:         ensure current_cov is positive definite
8:         mean_vector ← mean of continuous_data
9:         **for** each row $i$ in individual_data **do**
10:            **if** random value < mutation_prob **then**
11:                new_continuous ← sample from multivariate normal with mean_vector and current_cov
12:                mutated[$i$, continuous_idx] ← new_continuous
13:            **end if**
14:        **end for**
15:     **end if**
16:     **if** categorical_idx and cat_probs exist **then**
17:         **for** each row $i$ in individual_data **do**
18:            **if** random value < mutation_prob **then**
19:                **for** each categorical index $j$ **do**
20:                    mutated[$i$, cat_idx] ← sample from categorical distribution with probabilities cat_probs[$j$]
21:                **end for**
22:            **end if**
23:        **end for**
24:     **end if**
25:     mutated ← replace NaN values with 0
26:     round and clip categorical values to valid range
27:     **return** flatten mutated array to 1D list
28: **end function**

---

**Algorithm 7** Crossover

---

1: **function** CROSSOVER(ind1, ind2, cxpb, row_mode_prob, n_features)
2:     **if** random value $\geq$ cxpb **then**
3:         **return** ind1, ind2                                      ▷ No crossover performed
4:     **end if**
5:     matrix1 $\leftarrow$ reshape ind1 to 2D array with $n\_features$ columns
6:     matrix2 $\leftarrow$ reshape ind2 to 2D array with $n\_features$ columns
7:     $n\_samples \leftarrow$ number of rows in matrix1
8:     **if** random value $<$ row_mode_prob **then**
9:         perform row-wise crossover
10:         $n\_rows \leftarrow \lfloor 0.3 \times n\_samples \rfloor$
11:         select $n\_rows$ random row indices
12:         swap selected rows between matrix1 and matrix2
13:     **else**
14:         perform column-wise crossover
15:         $n\_cols \leftarrow \lfloor 0.3 \times n\_features \rfloor$
16:         select $n\_cols$ random column indices
17:         swap selected columns between matrix1 and matrix2
18:     **end if**
19:     matrix1 $\leftarrow$ replace NaN values with 0
20:     matrix2 $\leftarrow$ replace NaN values with 0
21:     ind1 $\leftarrow$ flatten matrix1 to 1D list
22:     ind2 $\leftarrow$ flatten matrix2 to 1D list
23:     **return** ind1, ind2
24: **end function**

---

# B   Selection of meta-features

The following experiment was conducted with the objective of selecting the most informative meta-features for the purpose of OOD evaluation. Such meta-features, which exhibit changes in their values that would accurately indicate a shift in the data, are of particular interest. The generation of toy-data with different types of shifts (synthetic) was undertaken, and then the changes in the values of meta-features (absolute differences) were measured on subsamples within the domain and between domains. Subsequently, a range of methodologies for meta-feature selection ([3]) were employed, and the frequency with which each meta-feature was selected was calculated (see Figure 2). The data with shifts were modeled synthetically; for example, the distribution $p(y|x)$ (concept shift) was explicitly altered, or the distributions of predictors (covariance shift) were explicitly changed.

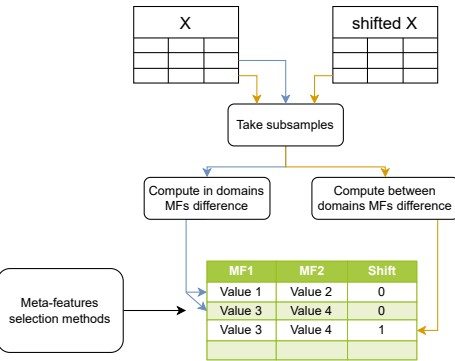

**Figure 2:** General outline of the experiment to determine the meta-features that respond best to the shift.

The selection of statistical and information-theoretical meta-characteristics as the primary groups was made on the basis of their relative simplicity in terms of interpretation. Furthermore, an investigation was conducted into various summarizing functions. Further information regarding meta-features and summarizing functions can be found in the following source ([24]). The figure 3 presents the concluding outcomes of the investigation into the variability of meta-characteristics across diverse groups under varying shifts.

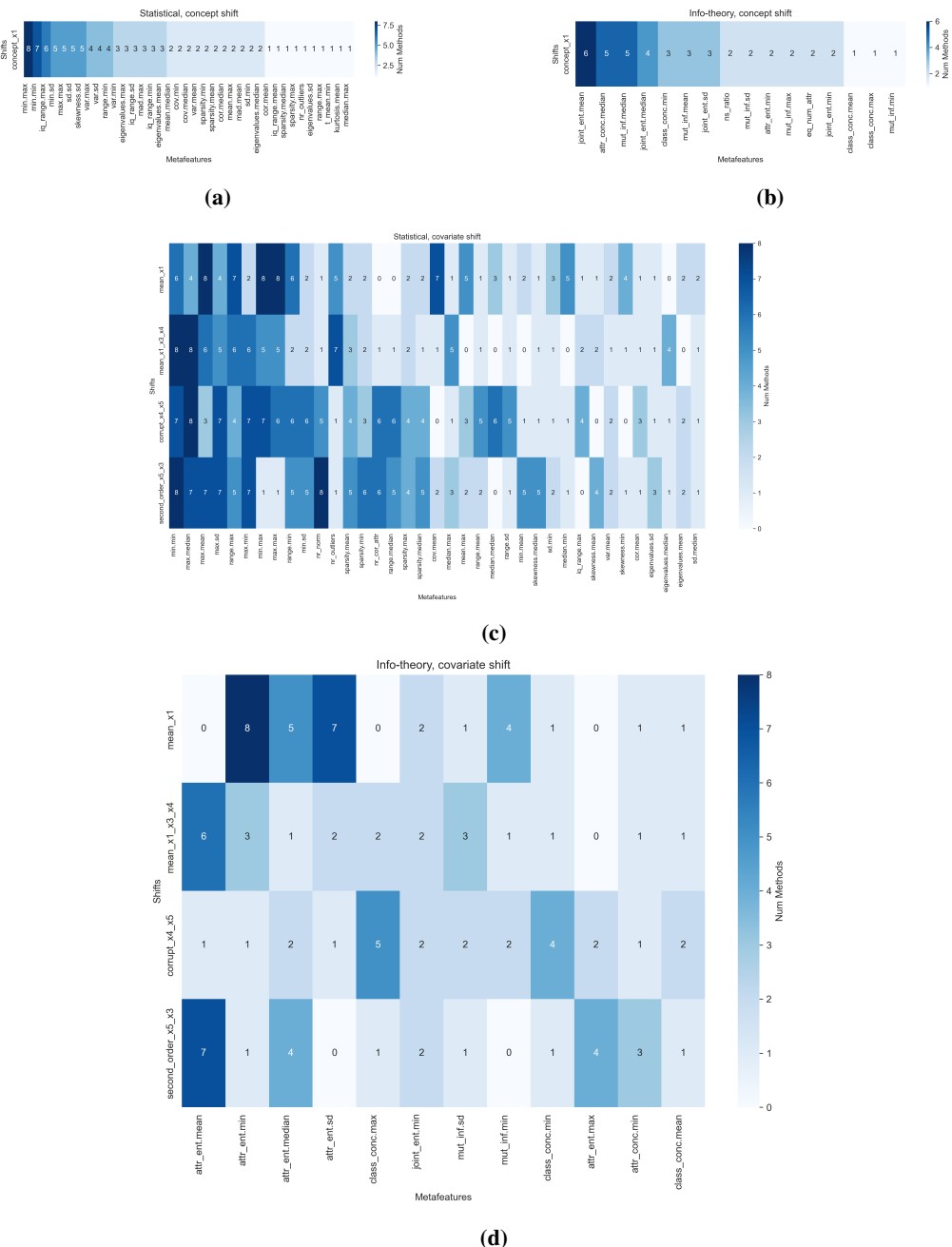

**Figure 3:** Results of the analysis of meta-feature variability under different shifts. Here, the meta-feature is named according to the principle $meta\_feature\_name.summarizing\_function$. The number on the diagram indicates how many meta-feature selection methods chose this meta-feature as significant for a given shift type.

# C  Generation of synthetic OOD data

The following graphs compare the values of meta-features on real data and synthetic data (figure 4 for california dataset and figure 5 for electricity dataset). It is important to note that a decision was taken not to aggregate the meta-feature vector in the generation task, as this resulted in substantial quality degradation. Consequently, each meta-feature is represented by a vector in this study.

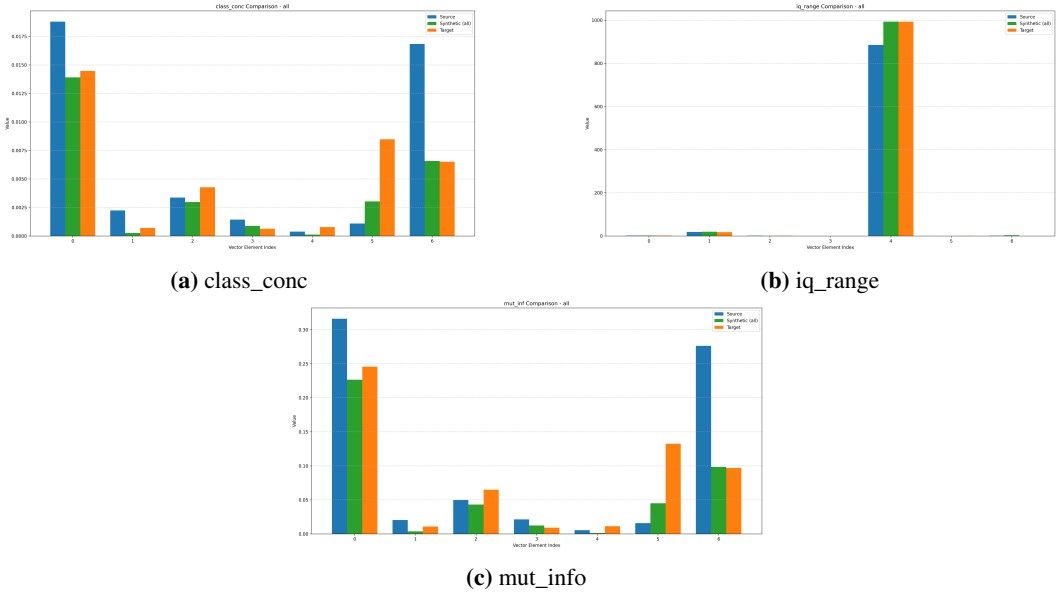

**Figure 4:** Comparison of meta-feature values on real and synthetic data for the dataset california.

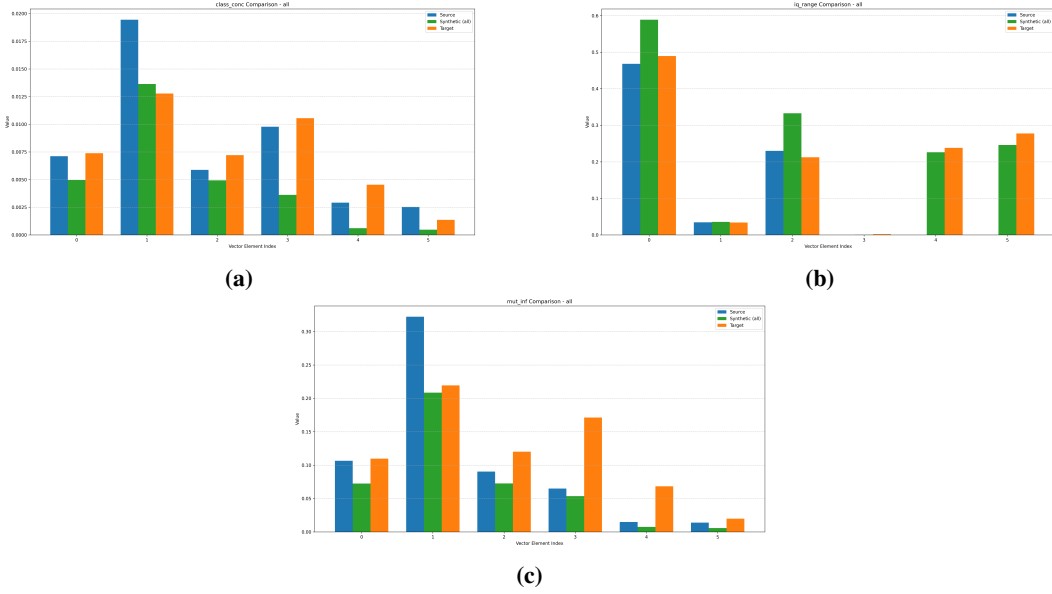

**Figure 5:** Comparison of meta-feature values on real and synthetic data for the dataset electricity.

