# OpenReview forum: "Evaluating robustness of tabular models under meta-features based shifts"
_NeurIPS.cc/2025/Workshop/Reliable_ML — NeurIPS 2025 - Reliable ML Workshop_

### Official Review · Reviewer_vQwK · 2025-09-13

**Rating:** 7
**Confidence:** 3

**Review:**

**Summary:**

The authors examine the problem of evaluating the out-of-distribution (OOD) performance of an ML model using only in-distribution samples available at training time. They propose two meta-feature based methods of simulating OOD samples for this purpose. The first is based on splitting the training dataset into two subsets that differ significantly on some user-defined meta-features, and using one subset for training and the other for evaluation. The second relies on using an evolutionary algorithm to generate a synthetic OOD dataset whose meta-feature values are close to some given target values. The authors then examine the effectiveness of their methods experimentally, comparing them to simply evaluating a model on a random split of the training data. They illustrate that evaluating on their simulated OOD dataset is more indicative of the model’s actual performance on OOD data, and also that the simulated datasets exhibit greater distributional differences from the initial dataset.

**Strengths:**

The research direction of simulating datasets that effectively predict a model’s OOD performance seems fruitful, and its connections with reliable ML are clear. The idea of using evolutionary algorithms (instead of more advanced or heavy-weight generative models) for this purpose is also interesting and appears effective. The experiments are presented in detail, and provide support for the effectiveness of the presented algorithms, as well as for the usefulness of the considered meta-features. Overall, the paper is easy to follow, while its ideas and methodologies are clearly explained.

**Weaknesses/Limitations:**

I am not particularly familiar with the field, however, it does seem to me that the paper is somewhat lacking in impact. Specifically, the benchmark used to evaluate the effectiveness of the OOD-generating algorithms is simply a random split of the data, which seems rather weak. It would be more compelling to compare the results to existing works (assuming such works exist), or even to implement more advanced algorithms/models and compare their effectiveness (though this is a limitation that the authors acknowledge).

I also feel that the mathematical notation could be more cleanly presented. There are a few instances where notation is used without first being defined, for example the meta-features $m_j$ in lines 106-107, as well as the feature-label pairs $X_i, Y$ and the probability mass functions $p(x_i, y), p(x_i), p(y)$ in lines 165-177.

**Suggestions for the Authors:**

It was not clear to me while reading how the target meta values $m_j^*$ referred to in lines 131 and 206-207 are computed. It was also not clear why the F1-score is the appropriate error metric for the experiments. Perhaps those points could be elucidated.

---

### Official Review · Reviewer_4UaT · 2025-09-20

**Rating:** 6
**Confidence:** 3

**Review:**

This paper introduces meta-feature–based splitting and synthetic generation as protocols for evaluating tabular models under distribution shift when true OOD data are unavailable. The approach is well motivated, clearly formalized, and supported by open-source code. Empirical results show that certain meta-features (mutual information, class concentration, joint entropy) indeed correlate with OOD degradation, and synthetic generation can approximate real targets when splitting fails.

However, the work has important limitations. The evaluation is restricted to five binary classification datasets with relatively simple shifts, leaving open whether the method generalizes to more diverse or high-stakes domains. The reliance on evolutionary optimization makes the protocol potentially expensive, yet compute costs are not reported. Baselines are weak—mostly random splits—while stronger alternatives (e.g., clustering-based splits, TabOOD) are not fully benchmarked. The synthetic generation component relies on ad-hoc mutations of a fixed generative prior, and its fidelity remains questionable compared to modern conditional generative models. Moreover, aggregation of meta-feature vectors hides per-feature variability, reducing interpretability of which predictors drive observed shifts.

Overall, the paper makes a useful contribution by highlighting meta-features as signals for robustness evaluation, but its scope, baseline comparisons, and methodological rigor fall short of a seminal advance. Strengthening baselines, reporting computational feasibility, and validating on more diverse tasks are needed before this can be considered a reliable protocol.